# The Effect of Different Colistin Dosing Regimens on Nephrotoxicity: A Cohort Study

**DOI:** 10.3390/antibiotics11081066

**Published:** 2022-08-05

**Authors:** Michael Samarkos, Konstantinos Papanikolaou, Athena Sourdi, Nikolaos Paisios, Efstratios Mainas, Elisabeth Paramythiotou, Anastasia Antoniadou, Helen Sambatakou, Panayiotis Gargalianos-Kakolyris, Athanasios Skoutelis, George L. Daikos

**Affiliations:** 11st Department of Medicine, Laikon General Hospital, Medical School, National and Kapodistrian University of Athens, 11527 Athens, Greece; 25th Department of Medicine, Evaggelismos Hospital, 10676 Athens, Greece; 31st Propaedeutic Department of Medicine, Laikon General Hospital, Medical School, National and Kapodistrian University of Athens, 11527 Athens, Greece; 41st Department of Medicine, G. Gennimatas General Hospital, 11527 Athens, Greece; 52nd Department of Medicine, Ippokrateion General Hospital, Medical School, National and Kapodistrian University of Athens, 11527 Athens, Greece; 6Intensive Care Unit, Attiko University Hospital, 12462 Athens, Greece; 74th Department of Medicine, Attikon University Hospital, Medical School, National and Kapodistrian University of Athens, 12462 Athens, Greece

**Keywords:** colistin, RIFLE, acute kidney injury, nephrotoxicity, pharmacokinetics/pharmacodynamics

## Abstract

(1) Background: It is not known whether different daily dosing schemes have different effects on colistin nephrotoxicity. We examined the effect of once- versus twice- or thrice-daily doses of colistin on renal function. (2) Methods: We performed a multicenter retrospective cohort study of hospitalized patients with a baseline glomerular filtration rate ≥ 50 mL/min who received intravenously the same colistin dose once (regimen A), twice (regimen B) or thrice daily (regimen C). The primary endpoint was acute kidney injury (AKI), defined as fulfilment of any of the RIFLE (Risk-Injury-Failure-Loss-End stage renal disease) criteria. (3) Results: We included 306 patients; 132 (43.1%) received regimen A, 151 (49.3%) regimen B, and 23 (7.5%) regimen C. Ninety-nine (32.4%) patients developed AKI; there was no difference between regimen A vs. B and C [45 (34.1%) vs. 54 (31.0%), *p* = 0.57]. In a propensity score–matched cohort, AKI was similar in patients receiving Regimen A, Regimen B, and Regimen C (31.6% vs. 33.3%, *p* = 0.78). On logistic regression analysis, diabetes was an independent predictor of AKI (OR = 4.59, 95% CI 2.03–10.39, *p* = 0.001) while eGFR > 80 mL/min (OR = 0.50, 95% CI 0.25–0.99, *p* = 0.048) was inversely associated with AKI. (4) Conclusions: Colistin once daily is not more nephrotoxic than the standard colistin regimens. The only independent predictor of nephrotoxicity was diabetes mellitus, while eGFR > 80 mL/min had a protective effect.

## 1. Introduction

Colistin (polymyxin E), a cationic polypeptide antibiotic, is active against extensively drug-resistant (XDR) organisms such as carbapenem-resistant *enterobacterales* (CRE), *Pseudomonas aeruginosa* (CRPA), and *Acinetobacter baumanii* (CRAB). Despite the recent development of several novel agents for the treatment of XDR Gram-negative infections, it can be reasonably expected that the use of colistin will continue, as novel agents are not active against all XDR pathogens and are not universally available [1].

One of the major disadvantages in the clinical use of colistin is the narrow therapeutic window and its association with nephrotoxicity. In three large meta-analyses, the rate ranged from 37.8% to 48% [2,3,4]. Advanced age, chronic comorbid conditions, hypoalbuminemia, and concomitant administration of other nephrotoxic agents are among the factors most commonly identified as predictors of nephrotoxicity [3,5,6] The potential relationship of different colistin dosing schemes with nephrotoxicity, however, has not been previously examined.

Over the last two decades, significant progress has been made to better understand the colistin kinetics and define the pharmacokinetic/pharmacodynamic (PK/PD) indices and dosing schemes best correlated with clinical outcome [7,8,9,10]. The free-drug area under the concentration–time curve to MIC ratio (*f*AUC/MIC) appears to be the PK/PD index best associated with efficacy [9]. To drive the PK/PD profile in adequate exposure, it is recommended to administer an initial loading dose of colistin followed by a maintenance dose divided into two daily dosages [9]. The currently used dosing schemes (initial loading dose of colistin followed by maintenance dose divided into two or three daily dosages), however, result in a plasma concentration of formed colistin which may not be effective in lower respiratory tract infections and in infections caused by pathogens with colistin MICs >0.5 mg/L [8,11,12]. Thus, searching for a more effective dosing scheme may be worth trying.

Several studies have indicated that colistin exerts a concentration-dependent killing against Gram-negative pathogens [13,14,15]. Also, in an in vitro PK/PD model, Tsala et al. showed that both free maximum drug concentration to MIC ratio (*f*Cmax/MIC) and the free-drug area under the concentration–time curve to MIC ratio (*f*AUC/MIC) were associated with bacterial killing and that dosing regimens attaining *f*Cmax/MIC ≥6 always resulted in bactericidal activity [12]. In addition, it has been demonstrated that trough plasma level of colistin is an independent predictor of nephrotoxicity [16]. These observations suggest that larger doses at longer intervals may optimize PK/PD indices and offer an advantage over twice or thrice daily doses in terms of efficacy and safety. Nevertheless, if any efforts are going to be made to enhance colistin efficacy by different dosing schemes, we need to ensure first that high dose-extended interval regimens are not more nephrotoxic [17].

Our hypothesis was that a once-daily dose of colistin is not more nephrotoxic than the twice- or thrice-daily dosing schemes. Thus, we examined the impact of different colistin dosing schemes (once-daily versus twice- or thrice-daily doses) on renal function and aimed to identify predictors of acute kidney injury (AKI)A using the RIFLE criteria (Risk-Injury-Failure-Loss-End stage renal disease) in patients with a baseline eGFR ≥50 mL/min.

## 2. Results

### 2.1. Patient Characteristics

Three hundred and six patients were enrolled in the study; 207 (67.6%) were males and 99 (32.4%) females, with a mean age of 57 ± 16.0 years. The majority of thepatients were Caucasians (301/306, 98.3%). One hundred thirteen (36.9%) had been hospitalized in surgical wards, 84 (27.5%) in medical, and 109 (35.6%) in ICU.

Regimen A was administered in 132 (43.1%), regimen B in 151 (49.3%), and regimen C in 23 (7.5%) patients. Loading dose was administered in 194 patients (63.4%). The duration of the colistin treatment was 16.9 ± 10.5 days and the mean cumulative colistin dose was 150.9 ± 90.9 MIU. Two hundred and thirty-six patients (77.8%) received concomitantly at least one potentially nephrotoxic agent, the most common category being glycopeptides. The most common infection in the medical patients was primary bacteraemia (23/84, 27.4%), followed by pneumonia (18/84, 21.4%). In the ICU patients the most common infection was VAP/Pneumonia (75/109, 68.8%), followed by primary bacteremia (24/109, 22.0%). In the surgical patients, abdominal infections predominated (43/113, 38.1%), followed by SSTIs (17/113, 15.0%). Details regarding the baseline characteristics of the full cohort and of the two comparison groups are shown in Table 1, while details per treatment regimen are shown in Appendix A.

### 2.2. Outcomes

As shown in Table 2, there were no significant differences in any of the primary or secondary outcomes between the treatment groups (regimen A vs. regimens B or C). AKI by RIFLE classification was noted in 45 (34.1%) of 132 patients treated with regimen A and in 54 (31.0%) of the 174 treated with regimen B or C. Colistin was discontinued due to AKI in 16 (12.1%) patients of regimen A and in 16 (9.2%) of regimen B or C. Reversal of eGFR to pre-treatment levels was noted in 18 (75.0%) of the 24 patients treated with regimen A and in 28 (68.3%) of the 41 patients of regimen B or C for whom three-month follow-up data were available. All-cause 30-day mortality was no different between treatment groups (14.6% for regimen A vs. 20.1% for regimen B and C).

### 2.3. Predictors of AKI—All Patients

Univariate analysis in the whole cohort of 306 patients revealed that age (*p* = 0.01), diabetes (*p* = 0.001), CCI (*p* = 0.001), and the concurrent use of at least one nephrotoxic agent (*p* = 0.04) were associated with AKI (no fulfilment of RIFLE criteria vs. any RIFLE), whereas patients with eGFR > 80 mL/min had a significantly lower risk (*p* = 0.02). Of note, no association was observed with dosing regimen A vs. regimens B and C, administration of loading dose, duration of treatment, or cumulative colistin dose (Table 3).

### 2.4. Predictors of AKI—Propensity Score–Matched Cohort

The PSM cohort included 117 patients of Regimen A and 117 patients of Regimens B and C. The variables included in the model were: age, ICU admission, severe sepsis or septic shock, CCI, diabetes, APACHE II score, co-administration of at least one potentially nephrotoxic drug, and baseline eGFR > 80 mL/min. The two groups did not differ significantly in terms of age, BMI, comorbidities, severity of infection (i.e., presence of septic shock, mechanical ventilation, haemodynamic instability) or in treatment-related variables (i.e., administration of loading dose, duration of treatment, and cumulative colistin dose).

AKI was similar in the two groups [Regimen A, 37/117 (31.6%) vs. Regimen B and C 39/117 (33.3%), *p* = 0.78]. Also, no differences were observed between the groups in the secondary outcomes: colistin discontinuation rate, reversal of AKI, all-cause 30-day mortality (see Table 2).

In univariate analysis, CCI (*p* = 0.011), diabetes (Cp < 0.001), baseline eGFR > 80 mL/min (*p* = 0.01), and mechanical ventilation (*p* = 0.046) were significantly associated with AKI, as defined by RIFLE criteria, whereas the colistin-dosing regimen had no association (Table 3). By entering the significantly associated variables of the PSM cohort, as well as severe sepsis or septic shock and co-administration of a nephrotoxic agent, as potential variables associated with AKI in a binary logistic regression model, diabetes [Odds ratio (OR) = 4.59, 95% CI 2.03–10.39, *p* = 0.001] remained an independent predictor of AKI while eGFR > 80 mL/min (OR = 0.50, 95% CI 0.25–0.99, *p* = 0.048) was inversely associated with AKI (Table 4).

## 3. Discussion

Herein was shown that AKI, as defined by RIFLE criteria, occurred in 99 (32.4%) of 306 patients who had received colistin treatment for a variety of infections. Our analysis could not establish an association between AKI and colistin dosing schemes or any other colistin treatment variables (loading dose, duration of treatment, or cumulative colistin dose). The only variable independently associated with AKI was the presence of diabetes (OR = 4.59), while eGFR > 80 mL/min was inversely associated with AKI (OR = 0.50).

Colistin nephrotoxicity is the most clinically significant dose-limiting adverse event. In our study, colistin-induced AKI was slightly lower than that reported in three large meta-analyses (37.8% to 48%) [2,3,4]. Also, the severity of AKI by RIFLE criteria (Risk = 12.4%, Injury = 12.4%, Failure = 7.5%) was lower than the 17%, 13%, and 10% for “Risk,” “Injury” and “Failure,” respectively, reported by Sisay et al. [3]. These differences could be attributed to differences in the study population and to the colistin dose used [4]. Our study participants had several distinct characteristics that could explain the lower incidence and severity of nephrotoxicity: (i) all the patients had preserved renal function, patients with eGFR <50 mL/min were excluded from the study, (ii) 78.4% of the patients had eGFR greater than 80 mL/min, (iii) all the study participants received the same total daily dose (9 MIU), and (iv) the dose did not increase when the eGFR was >80 mL/min as recommended by the current guidelines [2]. The identified risk factors in univariate analysis associated with nephrotoxicity, advanced age, diabetes, CCI, mechanical ventilation, and concomitant administration of at least one nephrotoxic agent largely conform to the ones reported in the literature [5,6]. After multivariable logistic regression analysis, only diabetes remained an independent predictor of nephrotoxicity, whereas eGFR >80 mL/min had a protective effect.

The primary aim of our study was to evaluate the impact of different colistin-dosing regimens on nephrotoxicity. Of note, no difference was observed in AKI with respect to dosing frequency, once-daily vs. multiple-daily doses, in the full cohort. Similar results were obtained in the propensity-matched cohort; AKI occurred in 31.6% of the patients who received colistin once-daily and in 33.3% of those who received the drug twice- or thrice-daily. Also, no differences were observed between the two dosing schemes in the secondary outcomes: the severity of nephrotoxicity, as defined by RIFLE criteria, the treatment discontinuation rate due to toxicity, the reversal of nephrotoxicity, and all-cause 30-day mortality.

Preliminary pharmacokinetic data of colistin support the notion that the once-daily dose scheme might be less nephrotoxic than the fractionated dosing regimens [16,18]. In a study by Margeault et al., the once-daily administration of 9MIU of colistin resulted in Cmax of 7.2 mg/L and in a relatively low trough concentration of 1.7 mg/L [18]. In the same study, in population pharmacokinetic analysis, it was estimated that 80% of the simulations with the once-daily scheme, on day 7 of the treatment, had trough colistin concentrations less than 3.33 mg/L, a value that has been shown to be an independent predictor of AKI [16]. In comparison, the twice- and thrice-daily dosing schemes were estimated to have drug concentrations of <3.3 mg/L in only 57% and 42% of the simulations, respectively [18]. On the other hand, there are data suggesting that peak colistin concentration may be a predictor of nephrotoxicity [19]. Provided that the once-daily dosing achieves higher peak concentrations than multiple-daily dosing, it is possible to be more nephrotoxic [20]. Our findings, however, did not support this hypothesis.

To our knowledge, this is the first study examining the effect of colistin-dosing schemes on nephrotoxicity in humans. Okoduwa et al. have examined the effect of once- vs. twice-daily dosing of polymyxin B on AKI in a propensity-matched cohort [21]. Contrary to our findings, the investigators reported that the once-daily regimen of polymyxin B was more likely to induce AKI than the twice-daily dose scheme [21]. It should be pointed out, however, that these findings should not be extrapolated to colistin as there are several pharmacologic differences between CMS/colistin and polymyxin B that could influence the results [22,23]. Opposite results to those obtained in humans have been observed in experimental infections. Wallace et al., using a rat model, found that the colistin-dosing regimen mimicking the once-daily dose in humans was significantly more nephrotoxic than the twice-daily scheme [19]. In contrast, other investigators, using polymyxin B in a rat model, demonstrated lower incidence of nephrotoxicity in the once-daily dose as compared to split doses [24].

Our findings should not be interpreted without considering several limitations. We acknowledge the retrospective character of the study and the inherent shortcomings that exist in this type of study. Thus, our results may well be driven by unrecognized variables besides the colistin-dosing schemes. However, propensity score matching was used to minimize the differences between the treatment groups, and logistic regression analysis was performed to control for potential confounders. Also, the study period (2012–2014) from which the data were collected may not reflect current practices with respect to the treatment of multidrug resistant infections. Nevertheless, during that time period we had the opportunity to compare different dosing schemes as several physicians were using the once-daily dose of colistin. Another limitation of the study was the lack of data on variables with a potential effect on nephrotoxicity such as concurrent administration of “nephroprotective” agents (e.g., antihypertensive drugs) or disease duration and compliance with therapy in diabetic subjects.

## 4. Patients and Methods

### 4.1. Study Design

This is a multicenter retrospective cohort study which included patients hospitalized from February 2012 to June 2014 in five tertiary teaching hospitals located in Athens, Greece. The study was approved by the Institutional Review Boards of the participating centers (Reg no 973/10-07-2013).

### 4.2. Patient Population (Participants)

We have included consecutive inpatients from the Medical and Surgical Wards as well as ICUs who were treated with colistin. Patients were eligible for the study if they were >18 years old with a baseline glomerular filtration rate ≥50 mL/min as estimated by the Cockcroft–Gault formula (eGFR) and had received intravenous (iv) colistimethate sodium (CMS) for at least 48 h (Colistin/Norma^®^, 1,000,000 IU/34 mg colistin base activity; Norma Hellas S.A., Athens, Greece). Patients were excluded if they had eGFR <50 mL/min or if they had received intermittent or continuous renal replacement therapy at any time during the colistin treatment.

All the patients had received the same total of 9 million IU of colistin daily in one (Regimen A), two (Regimen B), or three (Regimen C) divided doses. The choice of the number of daily doses (i.e., one or more) and the administration of a loading dose of 9 million IU of colistin was at the discretion of the treating physician. All patients of Regimen A were considered as having received a loading dose. The treating physician made all colistin dose adjustments necessary whenever a change in renal function occurred, according to the guidance at the time of the study. The decision to discontinue the colistin treatment was at the discretion of the treating physician, on the basis of the existing alternatives to colistin treatment options (if any), how many days of treatment the patient had already received, and the clinical course of the patient.

Patients were followed-up for nephrotoxicity with laboratory testing at least twice per week, and urine output was recorded daily. The treating physician could order more frequent laboratory testing depending on the clinical circumstances. There were no scheduled follow-up visits. The treating physician arranged for post-discharge visits as needed.

The data were abstracted from the medical records to a standard form. We recorded demographics, body weight and height measurements, co-morbidities and Charlson Comorbidity Index (CCI), pertinent laboratory findings (such as serum creatinine-SCr), pathogen (if isolated), and any potentially nephrotoxic co-administered drugs. Glycopeptides, aminoglycosides, liposomal amphotericin, intravenous contrast agents, loop diuretics, non-steroidal anti-inflammatory drugs, and nephrotoxic antineoplastic chemotherapy (e.g., cisplatin) were included in the “Concomitant nephrotoxic agents” group. Any other drug which can occasionally be nephrotoxic was included in the “Other nephrotoxic drugs” group. The source of infection was defined on the basis of clinical, laboratory, and imaging findings, as well as culture results. The severity of illness was assessed using the APACHE II score on the day of the colistin initiation and we used the 2001 SCCM/ESICM/ACCP/ATS/SIS International Sepsis Definitions Conference criteria [25].

### 4.3. Outcomes

The primary endpoint included acute kidney injury (AKI) defined as fulfilment of any of the RIFLE criteria [26] at the end of the treatment. Secondary endpoints included discontinuation of treatment because of AKI, reversal of AKI (defined as absence of any of the RIFLE criteria) within three months, and all-cause 30-day mortality.

### 4.4. Statistical Analysis

For comparisons, the patients were divided into two groups; those who received regimen A versus those who received regimens B or C (regimens B and C were combined, as they attain similar peak and trough concentrations and do not differ significantly in PK/PD) [20]. We have used the chi-square or the Fisher’s exact test to compare proportions and the Mann–Whitney test to compare continuous variables between the two groups. Logistic regression was performed to investigate factors associated with AKI. To eliminate confounding factors affecting the choice of the colistin-dosing regimen (i.e., one or multiple dosages), a subset of patients, matched by propensity score, was analyzed. The propensity score model included as predictors all variables significantly associated with both the colistin-dosing regimen and AKI. The model also included variables with a potential association with nephrotoxicity according to the literature [27]. Based on the propensity score, with the tolerance margin set at 0.1, the patients of regimen A were matched to those of regimen B or C (1:1 matching).

All statistical analyses were performed with IBM SPSS Statistics version 25 (IBM Corporation, Armonk, NY, USA).

## 5. Conclusions

The findings presented herein showed that the once-daily dose of colistin is not more nephrotoxic than the twice- or thrice-daily dosing schemes. These findings coupled with the results of other studies provide a basis to consider investigating further the administration of colistin in higher doses at longer intervals in terms of safety and efficacy [12,28].

## Figures and Tables

**Table 1 antibiotics-11-01066-t001:** Patient characteristics by dosing scheme in the full cohort.

Variable	All Patients (n = 306)	Regimen A (n = 132)	Regimen B or C (n = 174)	
	n, (%)	n, (%)	n, (%)	*p*
Patient Variables				
Gender, male	207, (67.6)	86, (65.2)	121, (69.5)	0.46
Age, years (median, IQR)	60, (47–69)	61, (48–70.5)	59, (47–68)	0.59
Weight, Kg (median, IQR)	73, (65–80)	75, (65–80)	72, (66–80)	0.70
BMI, (median, IQR)	25.0, (22.5–26.8)	24.8, (22.6–26.9)	25.2, (22.5–26.7)	0.72
Obesity (BMI > 30)	21, (6.9)	10, (7.6)	11, (6.4)	0.27
Ward				<0.001
Medical	84, (27.5)	28, (21.2)	56, (32.2)	0.033
Surgical	113, (36.9)	71, (53.8)	42, (24.1)	<0.001
ICU	109, (35.6)	33, (25.0)	76, (43.7)	0.001
Charlson Comorbidity Index > 3	135, (44.1)	63, (46.7)	72, (53.3)	0.26
APACHE score, (median, IQR)	11, (7–16)	10, (6–15)	12, (7–17)	0.005
Diabetes	48, (15.7)	15, (11.4)	33, (19.0)	0.07
Heart failure	32, (10.5)	10, (7.6)	22, (12.6)	0.151
Neutropenia (PMN < 500/μL)	30, (9.8)	11, (8.3)	19, (10.9)	0.45
Serum creatinine at day 0, mg/dL (median, IQR)	0.7, (0.5–0.9)	0.64, (0.5–0.83)	0.7, (0.5–0.9)	0.37
eGFR at baseline mL/min (median, IQR)	118.3, (84.7–148.0)	121.8, (83.1–147.1)	116.9, (85–148.3)	0.75
Baseline eGFR > 80 mL/min	240, (78.4)	104, (78.8)	136, (78.2)	0.97
Infection Variables				
Site of infection *				<0.001
Primary bacteraemia	55, (18.0)	14, (10.6)	41, (23.6)	0.003 *
UTI	19, (6.2)	7, (5.3)	12, (6.9)	ns
Pneumonia/VAP	112, (36.6)	44, (33.3)	68, (39.1)	ns
Abdominal infection	52, (17.0)	36, (27.3)	16, (9.2)	<0.001 *
SSTI	25, (8.2)	14, (10.6)	11, (6.3)	ns
Other sites	43 (20.9)	17 (12.9)	26 (14.5%)	
Pathogen *				0.14
*Acinetobacter* spp.	105, (34.3)	54, (40.9)	51, (29.3)	0.034 *
*Pseudomonas* spp.	39, (12.7)	11, (8.3)	28, (16.1)	0.044 *
*Klebsiella* spp.	70, (22.9)	30, (22.7)	40, (23.0)	ns
Other bacteria	11 (3.6)	5 (3.8)	6 (3.4)	
No bacteria isolated	81, (26.5)	32, (24.2)	49, (28.2)	ns
Septic shock	64, (20.9)	26, (19.7)	38, (21.8)	0.67
Hemodynamic instability	85, (27.8)	36, (27.3)	49, (28.2)	0.86
Mechanical Ventilation	107, (35.0)	33, (25.0)	74, (42.5)	0.001
Treatment Variables				
Empirical Treatment	91, (29.7)	33, (25.0)	58, (33.3)	0.13
Administration of loading dose	194, (63.4)	132, (100)	118, (67.8)	<0.001
Duration of treatment, days (median, IQR)	14, (10–22)	15, (9–23)	14, (10–21)	0.82
Colistin total dose, MU (median, IQR)	126, (90–189)	135, (90–198)	126, (81–180)	0.124
Concomitant nephrotoxic agents				
Diuretics	121, (39.5)	54, (40.9)	67, (38.5)	0.67
Aminoglycosides	50, (16.3)	27, (20.5)	23, (13.2)	0.09
Amphotericin	17, (5.6)	9, (6.8)	8, (4.6)	0.40
Glycopeptides	104, (34.0)	48, (36.4)	56, (32.2)	0.44
Chemotherapy	58, (19.0)	20, (15.2)	38, (21.8)	0.13
Radiocontrast Agents	88, (28.8)	47, (35.6)	41, (23.6)	0.021
Non-steroidal anti-inflammatory drug	36, (11.8)	18, (13.6)	18, (10.3)	0.37
Other nephrotoxic drugs	45, (14.7)	3, (2.3)	42, (24.1)	<0.001

eGFR: estimated glomerular filtration rate; IQR: Inter-quartile range; MU: Million units; SSTI: Skin and soft tissue infection; UTI: Urinary tract infections; VAP: ventilator-associated pneumonia. * Tests are adjusted for all pairwise comparisons using the Bonferroni correction.

**Table 2 antibiotics-11-01066-t002:** Outcomes in all patients and in the propensity-matched cohort.

Variable	Full Cohort				Propensity Matched Cohort			
	All Patients (n = 306)	Regimen A (n = 132)	Regimen B or C (n = 174)		All Patients (n = 234)	Regimen A (n = 117)	Regimen B or C (n = 117)	
	n, (%)	n, (%)	n, (%)	*p*	n, (%)	n, (%)	n, (%)	*p*
RIFLE Nephrotoxicity (n, %)	99, (32.4)	45, (34.1)	54, (31.0)	0.57	76, (32.5)	37, (31.6)	39, (33.3)	0.78
No RIFLE	207, (67.6)	87, (65.9)	120, (69.0)	0.43 *	158, (67.5	80, (68.4)	78, (66.7)	0.94 *
Risk	38, (12.4)	14, (10.6)	24, (13.8)		29, (12.4)	11, (9.4)	18, (15.4)	
Injury	38, (12.4)	20, (15.2)	18, (10.3)		32, (13.7)	19, (16.2)	13, (11.1)	
Failure	23, (7.5)	11, (8.3)	12, (6.9)		15, (6.4)	7, (6.0)	8, (6.8)	
Treatment discontinuation because of AKI	32, (10.5)	16, (12.1)	16, (9.2)	0.40	27, (11.5)	15, (12.8)	12, (10.3)	0.68
Reversal of AKI ^§^	46/65, (70.8)	18/24, (75.0)	28/41, (68.3)	0.56	35/48, (72.9)	14/18, (77.8)	21/30, (70)	0.56
Day 30 mortality	54, (17.6)	19, (14.6)	35, (20.1)	0.21	39, (16.7)	16, (13.9)	23, (19.7)	0.24

AKI = acute kidney injury; * Tests are adjusted for all pairwise comparisons using the Bonferroni correction (SPSS note). ^§^ Data for reversal of AKI 3 months after discontinuation of colistin treatment were available for 65 patients in the whole cohort of patients and for 48 patients in the propensity-matched cohort.

**Table 3 antibiotics-11-01066-t003:** Risk factors for nephrotoxicity–univariate analysis.

Variable	Full Cohort			Propensity Score Matched Cohort		
	No RIFLE (n = 207)	Any RIFLE (n = 99)		No RIFLE (n = 158)	Any RIFLE (n = 76)	
	n, %	n, %	*p*	n, %	n, %	*p*
Patient Variables						
Gender, male	137, (66.2)	70, (70.7)	0.43	102, (64.6)	51, (67.1)	0.70
Age, years (median, IQR)	59, (44–68)	62, (51–74)	0.011	60, (44–68)		0.06
Weight, Kg (median, IQR)	74.5, (65–80)	72, (65–80)	0.59	75, (65–80)		0.39
BMI, (median, IQR)	24.9, (22.4–26.9)	25.6, (22.5–26.6)	0.88	24.8, (22.1–26.6)		0.80
BMI Classification, Obese	14, (6.8)	7, (7.1)	0.67	9, (59.7)	5, (6.6)	0.36
ICU Admission	81, (39.1)	28, (28.3)	0.06	49, (31.0)	17, (22.4)	0.17
Diabetes	23, (11.1)	25, (25.3)	0.001	11, (7.0)	21, (27.6)	<0.001
Heart_Failure	18, (8.7)	14, (14.1)	0.14	9, (5.7)	9, (11.8)	0.10
Neutropenia (PMN < 500/μL)	19, (9.2)	11, (11.1)	0.59	14, (8.9)	11, (14.5)	0.19
Charlson Comorbidity Index > 3	87, (42.0)	48, (48.5)	0.29	68, (43.0)	40, (52.6)	0.16
Charlson Comorbidity Index (median, IQR)	3, (1–4)	3, (3–6)	0.001	10, (6–15)	9.5, (6.5–14)	0.001
APACHE score, (median, IQR)	11, (7–16)	10, (7–16)	0.91	3, (2–5)	5, (2–6)	0.45
Serum creatinine at day 0, mg/dL (median, IQR)	0.62, (0.5–0.9)	0.71, (0.52–0.9)	0.26	0.69, (0.51–0.9)	0.79, (0.56–0.9)	0.23
eGFR at baseline ml/min (median, IQR)	122.2, (87.4–156.6)	112.5, (75.1–137.7)	0.03	118.21, (86.11–147.96)	104.26, (73.35–134.11)	0.02
Baseline eGFR > 80 mL/min	170, (82.5)	70, (70.7)	0.02	129, (81.6)	51, (67.1)	0.01
Infection Variables						
Site of infection *			0.26			0.31
Primary bacteremia	35, (16.9)	20, (20.2)	ns	19, (12.0)	13, (17.1)	ns
UTI	11, (5.3)	8, (8.1)	ns	8, (5.1)	5, (6.6)	ns
Pneumoniae, VAP	78, (37.7)	34, (34.3)	ns	60, (38.0)	23, (30.3)	ns
Abdominal infection	33, (15.9)	19, (19.2)	ns	30, (19.0)	18, (23.7)	ns
SSTI	16, (7.7)	9, (9.1)	ns	12, (7.6)	9, (11.8)	ns
Pathogen (n, %) *			0.50			0.42
*Acinetobacter* spp.	71, (34.3)	34, (34.3)	ns	54, (34.2)	28, (36.8)	ns
*Pseudomonas* spp.	23, (11.1)	16, (16.2)	ns	17, (10.8)	9, (11.8)	ns
*Klebsiella* spp.	45, (21.7)	25, (25.3)	ns	31, (19.6)	21, (27.6)	ns
No pathogen isolated	60, (29.0)	21, (21.2)	ns	49, (31.0)	16, (21.1)	ns
Septic shock	40, (19.3)	24, (24.2)	0.32	27, (17.1)	13, (17.1)	1.00
Hemodynamic instability	55, (26.6)	30, (30.3)	0.49	41, (25.9)	19, (25.0)	0.88
Mechanical Ventilation	80, (38.6)	27, (27.3)	0.051	51, (32.3)	15, (19.7)	0.05
Treatment Variables						
Empirical Treatment	65, (31.4)	26, (26.3)	0.35	51, (32.3)	18, (23.7)	0.18
Once daily dosing	87, (42)	45, (45.5)	0.57	80, (50.6)	37, (48.7)	0.78
Loading dose	170, (82.1)	80, (80.8)	0.78	130, (82.3)	62, (81.6)	0.89
Duration of treatment, days (median, IQR)	15, (10–22)	14, (9–21)	0.49	15, (9–21)	14, (8–20)	0.46
Colistin total dose, MU (median, IQR)	135, (90–198)	126, (81–180)	0.12	135, (90–189)	117, (80–177)	0.07
Concomitant nephrotoxic agents						
Diuretics	75, (36.2)	46, (46.5)	0.09	54, (34.2)	34, (44.7)	0.07
Aminoglycosides	32, (15.5)	18, (18.2)	0.54	27, (17.1)	14, (18.4)	0.80
Amphotericin	9, (4.3)	8, (8.1)	0.18	8, (5.1)	8, (10.5)	0.07
Glycopeptides	70, (33.8)	34, (34.3)	0.93	59, (37.3)	26, (34.2)	0.64
Nephrotoxic Chemotherapy	38, (18.4)	20, (20.2)	0.70	29, (18.4)	15, (19.7)	0.80
Radiocontrast_Agents	57, (27.5)	31, (31.3)	0.49	41, (25.9)	23, (30.3)	0.49
NSAIDS	25, (12.1)	11, (11.1)	0.81	16, (10.1)	9, (11.8)	0.69
Other nephrotoxic drugs	25, (12.1)	20, (20.2)	0.06	21, (13.3)	16, (21.1)	0.13
At least 1 concomitant nephrotoxic drug	154, (74.4)	84, (84.8)	0.04	120, (75.9)	64, (84.2)	0.15

* Tests are adjusted for all pairwise comparisons using the Bonferroni correction.

**Table 4 antibiotics-11-01066-t004:** Multivariable logistic regression analysis of risk factors for nephrotoxicity (RIFLE)–Propensity matched cohort (n = 234).

Variable	*p*	Odds Ratio	95% C.I.
Once daily dosing	0.58	1.18	0.65–2.13
Presence of severe sepsis OR septic shock	0.28	1.46	0.73–2.95
Mechanical ventilation	0.1	0.53	0.24–1.14
Diabetes	<0.001	4.56	2.01–10.34
At least 1 concomitant nephrotoxic drug	0.16	1.74	0.80–3.78
eGFR > 80 mL/min	0.047	0.50	0.25–0.99
Charlson Comorbidity Index > 3	0.969	0.99	0.53–1.84

## Data Availability

The study data are available on request.

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
