# Peer review of "The Effect of Different Colistin Dosing Regimens on Nephrotoxicity: A Cohort Study"

_antibiotics, 2022, doi:10.3390/antibiotics11081066_

Round 1

Reviewer 1 Report

The Authors to their best ability have addressed the issue of colistin dosing regimens by analyzing more than three hundreds patient. The Paper presentation is good, analyses are clear and sound. How why the regimens B and C were combined in statistical analysis? In such way it is hard to know to which dosing regimen (B or C) regimen A is beneficial or at risk ? is the twice-daily or thrice-daily more efficient than once-daily in terms of low risk of nephrotoxicity?

Over all, the paper is of great importance.

Author Response

We thank reviewers for their constructive comments. All of their comments were taken into consideration and the manuscript was revised accordingly.

REVIEWER No 1

Comments and Suggestions for Authors

The Authors to their best ability have addressed the issue of colistin dosing regimens by analyzing more than three hundreds patient. The Paper presentation is good, analyses are clear and sound. How why the regimens B and C were combined in statistical analysis? In such way it is hard to know to which dosing regimen (B or C) regimen A is beneficial or at risk ? is the twice-daily or thrice-daily more efficient than once-daily in terms of low risk of nephrotoxicity?

Over all, the paper is of great importance.

We have compared regimen A (once daily) against regimens B and C combined, as regimens B and C attain similar peaks and troughs and do not differ significantly in PK/PD (ref 20). Also, separate statistical analysis would be limited by the small number of patients which received regimen C (23 patients). We have added a relevant comment in the Methods and we provide a supplementary table with the characteristics of patients for dosing regimen A, B and C separately.

Reviewer 2 Report

The authors should consider the followings:

1.          The authors should provide any nephroprotective drugs administered of the subjects, such as the monitoring drugs of hypertension, and/or some diabetic medication.

2.          In patient characteristics, please specify the % ethnicity of the recruited patient.

3.          Please specify the test(s) or diagnosis of diabetes used in this study, and any details of regarding the onset time of diabetes of the patients.

4.          As of Table 1, please specific and list the “concomitant nephrotoxic agents” and “Other nephrotoxic drugs” in details (as in a paragraph, or in supplementary information).

5.          The authors should clearly mention the novel findings of the research, in the part of abstract and conclusion.

6.          Please tabulate Regimen B and C, individually, or list the justification of the rationale(s) of presenting B (or C) together.

7.          The authors may expand the summary of the pharmacokinetics of colistin.

8.          The authors should indicate the reference number of the study regarding the approval of the Institutional Review Boards.

9.          The authors may increase the number of relevant references in the articles.

10.      The authors should correct the typo at Line 28, Wee to We.

11.      Please check carefully the spelling of the use of English across the article, such as Chemoterapy (in Table 1).

12.      In Methods, the authors should list the manufacturer name of colistin (as well as the batch number administered), and the pharmaceutical form and/or strength used.

Author Response

We thank reviewers for their constructive comments. All of their comments were taken into consideration and the manuscript was revised accordingly.

REVIEWER No 2

Comments and Suggestions for Authors

The authors should consider the followings:

  1. The authors should provide any nephroprotective drugs administered of the subjects, such as the monitoring drugs of hypertension, and/or some diabetic medication.
  • Unfortunately, we have no data regarding antihypertensive and antidiabetic medications of the patients. We have included this in the limitations of the study.
  1. In patient characteristics, please specify the % ethnicity of the recruited patient.
  • 301/306 (98.3%) of patients were Caucasians. There were 4 patients of Indian descent and one African patient. We have included this information in the results.
  1. Please specify the test(s) or diagnosis of diabetes used in this study, and any details of regarding the onset time of diabetes of the patients.
  • We had no new onset diabetes in our patients. All patients with diabetes had been already diagnosed with the standard methods. We feel that there is no need for specific mention of this. Unfortunately, we have no details regarding duration of diabetes in our patients. We have, therefore, included this in the limitations of the study.
  1. As of Table 1, please specific and list the “concomitant nephrotoxic agents” and “Other nephrotoxic drugs” in details (as in a paragraph, or in supplementary information).
  • “Concomitant nephrotoxic agents” in Table 1 is not a variable, but a subheading for the lines that follow [i.e., aminoglycosides, amphotericin, glycopeptides, diuretics, iv contrast agents, nephrotoxic chemotherapy (e.g., cisplatin), non-steroidal anti-inflammatory drugs]. We have listed these agents in the methods section.
  • Other potentially nephrotoxic drug: In this category we included any drug which can occasionally be nephrotoxic, such as e.g., ACE inhibitors, acyclovir, quinolones etc. We had no predefined list of these agents.
  1. The authors should clearly mention the novel findings of the research, in the part of abstract and conclusion.
  • We feel that the “Conclusions” of the Abstract (Lines 39-40 in the original submission) state clearly our novel findings (e.g., once daily colistin is not more nephrotoxic and eGFR>80 ml/min protects against colistin nephrotoxicity.).
  • Our findings are summarized in the last paragraph of the Discussion (L 220-24). We have moved this paragraph at section “5. Conclusions” so that it would be clearer.
  1. Please tabulate Regimen B and C, individually, or list the justification of the rationale(s) of presenting B (or C) together.
  • We have compared regimen A (once daily) against regimens B and C combined, as regimens B and C attain similar peaks and troughs and do not differ significantly in PK/PD (ref 20). Also, separate statistical analysis would be limited by the small number of patients which received regimen C (23 patients). We have added a relevant comment in the Methods and we provide a supplementary table with the characteristics of patients for dosing regimen A, B and C separately.
  1. The authors may expand the summary of the pharmacokinetics of colistin.
  • We have added a paragraph in the introduction.
  1. The authors should indicate the reference number of the study regarding the approval of the Institutional Review Boards.
  • We have included the Reference number of IRB approval in the methods
  1. The authors may increase the number of relevant references in the articles.
  • We have added the following references.
  • Luque, S.; Grau, S.; Valle, M.; Sorli, L.; Horcajada, J.P.; Segura, C.; Alvarez-Lerma, F. Differences in pharmacokinetics and pharmacodynamics of colistimethate sodium (CMS) and colistin between three different CMS dosage regimens in a critically ill patient infected by a multidrug-resistant Acinetobacter baumannii. Int J Antimicrob Agents 2013, 42, 178-181, doi:10.1016/j.ijantimicag.2013.04.018.
  • Poudyal, A.; Howden, B.P.; Bell, J.M.; Gao, W.; Owen, R.J.; Turnidge, J.D.; Nation, R.L.; Li, J. In vitro pharmacodynamics of colistin against multidrug-resistant Klebsiella pneumoniae. J Antimicrob Chemother 2008, 62, 1311-1318, doi:10.1093/jac/dkn425.
  • Zabidi, M.S.; Abu Bakar, R.; Musa, N.; Mustafa, S.; Wan Yusuf, W.N. Population Pharmacokinetics of Colistin Methanesulfonate Sodium and Colistin in Critically Ill Patients: A Systematic Review. Pharmaceuticals (Basel) 2021, 14, doi:10.3390/ph14090903.
  • Wallace, S.J.; Li, J.; Nation, R.L.; Rayner, C.R.; Taylor, D.; Middleton, D.; Milne, R.W.; Coulthard, K.; Turnidge, J.D. Subacute toxicity of colistin methanesulfonate in rats: comparison of various intravenous dosage regimens. Antimicrob Agents Chemother 2008, 52, 1159-1161, doi:10.1128/AAC.01101-07.
  • Couet, W.; Gregoire, N.; Marchand, S.; Mimoz, O. Colistin pharmacokinetics: the fog is lifting. Clin Microbiol Infect 2012, 18, 30-39, doi:10.1111/j.1469-0691.2011.03667.x
  1. The authors should correct the typo at Line 28, Wee to We.
  • Corrected
  1. Please check carefully the spelling of the use of English across the article, such as Chemoterapy (in Table 1).
  • We have corrected any spelling errors.
  1. In Methods, the authors should list the manufacturer name of colistin (as well as the batch number administered), and the pharmaceutical form and/or strength used.
  • We have included this information in the Methods.

Reviewer 3 Report

This study is fascinating for infection disease clinicians presently and worth publishing. In this manuscript was written in good scientific English language and style. The authors presented the introduction part and rationale of the study. I will suggest that the authors can include an explanation for the following in detail:

1.       Is it possible to explain the type of disease disorders in ICU and surgical patients?

2.       This manuscript does not discuss the PK/PD of Colistin.

3.       Effect of Obesity on nephrotoxicity

4.       In discussion authors should compare the outcome of the current study with previously published PK studies that predicted the risk of nephrotoxicity

5.       Any observation based on augmented renal clearance on nephrotoxicity?

Author Response

We thank reviewers for their constructive comments. All of their comments were taken into consideration and the manuscript was revised accordingly.

REVIEWER No 3

Comments and Suggestions for Authors

This study is fascinating for infection disease clinicians presently and worth publishing. In this manuscript was written in good scientific English language and style. The authors presented the introduction part and rationale of the study. I will suggest that the authors can include an explanation for the following in detail:

  1. Is it possible to explain the type of disease disorders in ICU and surgical patients?
  • In ICU patients the most common infection was VAP/Pneumonia (68.8%) followed by primary bacteraemia (22.0%). In surgical patients, abdominal infections predominated (38.1%) followed by SSTIs (15.0%). We have included this information in the text of the Results.
  1. This manuscript does not discuss the PK/PD of Colistin.
  • We have added a paragraph in the introduction.
  1. Effect of Obesity on nephrotoxicity.
  • Obesity as risk factor for nephrotoxicity has analysed in two ways: BMI as a continuous variable and Obesity (BMI>30) as a nominal variable (See Table 3).
  1. In discussion authors should compare the outcome of the current study with previously published PK studies that predicted the risk of nephrotoxicity.
  • We discuss the predicted risk of nephrotoxicity in relation to the dosing scheme in L180-9 of the original submission. We have expanded this paragraph with a relevant comment.
  1. Any observation based on augmented renal clearance on nephrotoxicity?
  • Among our findings in the logistic regression was that eGFR >80 ml/min was protective against nephrotoxicity, with an OR of 0.5 (See text and Table 4).